# Social Innovation and Food Provisioning during Covid-19: The Case of Urban–Rural Initiatives in the Province of Naples

**Valentina Cattivelli** [1],* and **Vincenzo Rusciano** [2]

1    Eurac Research, 39100 Bolzano/Bozen, Italy
2    Department of Economic & Legal Studies, "Parthenope" University, 80132 Naples, Italy;
     vincenzo.rusciano@uniparthenope.it
*    Correspondence: valentina.cattivelli@eurac.edu or valentina.cattivelli13@gmail.com

**Abstract:** This paper draws on the theoretical framework based on social innovation determinants to analyze how and to what extent the recent and self-organized initiatives for food provisioning are contributing to increase food accessibility at the time of personal and mobility restrictions due to Covid-19. Based on this, the paper firstly maps the initiatives activated during the first months of the Covid-19 emergency (March and April 2020) in the urban–rural territories in the province of Naples (Italy). Secondly, it characterizes these initiatives in relation to their capacity to enhance outcome and social well-being, as well as to involve local society in answer to social challenges through a desk research. Thirdly, the paper describes the case of Masseria Ferraioli, which emerges as social innovative best practice among the previous mapped initiatives. Even in these days, the Masseria distributes to people who cannot afford the purchase due to the emergency vegetables grown on land confiscated from the Camorra, the local mafia. Its configuration as social innovative experience is also confirmed directly by the Masseria´s project manager, who was required to answer to a semi-structured interview. Based on the evidence of the desk research and this interview, the paper demonstrates the importance of the combined commitment of local communities and volunteering association as a reaction to food provisioning problems in the time of Covid-19, as well as an increasing interest in reconnecting with local food practices, above all when food access has become a priority.

**Keywords:** social innovation; Naples; food provisioning; Masseria Ferraioli, Italy

## 1. Introduction

In recent years, there has been an increase in people needing food assistance in Europe [1,2]. Their condition is not a matter of availability of food, but a consequence of income disparities, economic downturn, and unemployment [3]. Their number will probably increase in the next months due to the Covid-19 diffusion. This exceptional event has undoubtedly direct and indirect effects on food provision, although to date it seems difficult to estimate their magnitude. The first and immediate effects could be on the practices in agricultural production within the food system, which are at risk due to the reduction of seasonal agricultural personnel. The second ones could interest non-food-related socio-economic and ecological drivers that constitute the context for food systems activities (i.e., processing, packaging, retailing and distribution and consuming) and might be affected by the restrictions imposed to local population and firms.

The literature on social innovation suggests possible solutions to address problems connected to the food provision. Social innovation theory explains the processes of innovation in social relations especially with regard to the governance [4]. As such, it refers to not only "particular actions, but also

to the mobilization-participation processes and to the outcome of actions which lead to improvements in social relations" [4] (p. 1). These actions take place in disadvantaged contexts, with few services and high risk of social isolation and exclusion, and they are launched and carried out by the local population. Technologies can be part of these processes; however, they do not play an essential role. The focus is rather placed on the self-organization of local population who promotes bottom-up approaches to overcome the lack of service provision. Evidence suggests that some social innovation initiatives in food provisioning were experienced in support of people needing of food assistance. The setting up shops (as in Bornholm, in Denmark), reorganizing AFN (alternative food networks) or their distribution network are some examples. These experiences in fact highlight the ability of local communities to develop local food networks to respond to locally identified problems (difficulties in food provision) towards a more sustainable change (fostering more sustainable production/distribution practices and individual behavior) [5–7].

In this time of restrictions imposed to prevent the spread of the pandemic, solidarity in the territories is increasing, reshaping local actions, mobilization and fostering self-organized initiatives in support of those in need. This is happened also in Naples province, which is one of the southern regions most affected by the spread of Covid-19. From the beginning of the health emergency, this province has distinguished itself for the proliferation of social innovation initiatives, including in the field of food provisioning. This province is also very exposed to the economic effects of the emergence of the pandemic: according to a Bank of Italy investigation, it is lagging behind in development.

This paper has two objectives. The first objective is to present an original and up-to-date characterization of the self-organized initiatives for food provision in the urban-rural territories in the province of Naples. Since the first days of the Covid-19 emergency, farms, voluntary associations, gardeners, canteens, churches and small farmers have been active to distribute food to disadvantaged people or people at risk of social and physical isolation.

The second objective of the paper is to explore the case of Masseria Ferraioli and to test its configuration as social innovation experience according to the findings of SIMRA project (Social innovation in marginal rural areas, H2020 project).

The structure of the article is organized as follows. The second section profiles the social innovation theory, according to the most recent studies and specifically to SIMRA project. The third section introduces some problems related to the current situation induced by Covid -19 diffusion. The following sections describe the test area and the method adopted within the present study, which mixes desk research analysis and semi-structured interview processing. The following section presents a map of food provisioning self-organized initiatives in the urban-rural territories in the Naples province and offers their characterization according to the four pillars of social innovation initiatives identified by SIMRA project. Subsequently, the next sections present the Masseria Ferraioli project and the findings of a semi-structured interview, which was submitted to its project manager in order to explore the coherence of the project with social innovation requirements. Finally, the last section draws the conclusions.

## 2. Theoretical Background

Spatial and income disparities complicate an equal access to services of general interest, as fundamental rights of all citizens and as a sign of democracy [8], as well as to opportunities, jobs and knowledge. The unfair or not equitable spatial distribution of services or the scarce opportunity to access them may create locational discrimination, (i.e., a discrimination imposed on certain population due to their geographical location) or income discrimination (i.e., a discrimination based on its income level [9]). These forms of discrimination are socially undesirable and unsustainable as they lead to a certain form of segregation and adversely affect the conditions essential to the normal course of human life [10,11].

Social innovation has attracted relevant academic interest in the last few years as a possible solution to counteract market and state failures, enabling local communities to develop new public services

or to maintain them [12–14]. This type of innovation is experienced in places where local population suffers from the lack of public services due to the geographical characteristics of the places they live in (remoteness, limited transport connections, and mountain or island morphological characteristics) or from exceptional conditions which lead to problems in maintaining the services provision.

Several studies have explored the characteristics of social innovation initiatives and theorized their profile [15,16]. Within the TEPSIE project (Theoretical, Empirical and Policy Foundations for Social Innovation in Europe, Horizon 2020 project [17]), social innovation is defined as a set of "new solutions (products, services, models, markets, processes etc.) that simultaneously meet a social need (more effectively than existing solutions) and lead to new or improved capabilities and better use of assets and resources. In other words, social innovation is both good for society and enhances society's capacity to act." Accordingly, social innovation experiences offer new solutions, which consist in products and services, but also in new models and processes. Doing so, they meet some needs with a pronounced social connotation more efficiently than other existing initiatives and improve capabilities and relationships among actors. Bock extends the concept of social innovation, reinforcing its social connotation: "Social innovation is a complex and multi-dimensional concept that is used to indicate the social mechanism, social objectives, and/or societal scope of innovation" (p. 58). Based on this, the author emphasizes the strong social connotation of the initiatives and process phases, which lead to their implementation, as well as he assumes that this concept may affect society as a whole and not just the singular individuals [18].

The most complete definition of social innovation derives from SIMRA project. This definition demonstrates the importance of social connotations, but, at the same time, it underlines the essential contribution of and for local communities. According to this project, social innovation is "the reconfiguring of social practices, in response to societal challenges, which seeks to enhance outcomes on societal well-being and necessarily includes the engagement of civil society actors" (SIMRA project, Horizon 2020 [19]). Therefore, its pillars are: (1) the reconfiguration of social practices, (2) the existence of societal challenges for which social innovation initiatives try to give solutions, (3) the attempts to enhance outcome and social well-being, (4) the engagement of society. In other terms, to be defined as social innovative, experiences should reconfigure existing social practices and give an answer to verified societal challenges. Doing so, they increase local outcome and well-being and involve directly local population. Acting in this way, social innovation initiatives imply a new philosophy of social interventions: the local population no longer relies just on the welfare state, and on the solutions developed by the central government, but realize that they themselves are the key to improve their own situation.

In recent times, social innovation initiatives have arisen in a vast number of contexts and sectors (healthcare [6,7], education [20,21], housing services [22], etc.) and relate to the management of service supply or its content. There is a growing trend also in food provisioning initiatives and specifically for those related to sustainable and social agriculture, locally oriented and alternative to the traditional food access channels. Initiatives like CSA (Community Supported Agriculture), urban gardening, consumer purchasing groups and other AFN (Alternative food network) initiatives reassess the networks through which food passes and encourage the sense the responsibility and the participation of local community. As example, Signori and Forno [5] apply the social innovation scheme to solidarity purchasing groups (GAS—Gruppo di acquisto solidale) in Italy. The authors detect that the mechanisms and processes underlying the purchasing sustain such forms of collective action and perform the sense of social effectiveness through the involvement of local population. This increases the capacity of communities to respond to locally identified problems. Reconfiguring their purchasing practices, interviewed people also reveal themselves to be more careful to local environmental sustainability and the need to improve it, as well as to recognize the role of socialization in both individual learning and local empowerment. Referring to Bornholm, a small island in Denmark, ESPON Bridges project [23] provides an example of how civil society can mobilize to revitalize a place while raising the quality of life of local population, thereby assisting in the provision of services of general interest and specifically in food provision.

In this Danish island, due to the absence of any shop, local population promotes the start-up of a community-driven shop in response to a failure of local authorities to provide services in food areas that are needed. The shop has quickly become a place for elderlies to meet and exchange and for community events thus bringing further social benefits for the whole community. Exploring the organic food buying groups in Valencia, Sifres et al. [24] point out on one hand the complexity, richness and specificity of bottom-up processes of innovation; on the other, the specific contributions of this process to social transformation. In the middle, they emphasize the relevance of people-driven processes in promoting people's ability to configure, plan and carry out concrete actions.

## 3. The Current Situation: Covid-19, Restrictions and Problems in Food Access

In recent weeks, the theme of food access has been debated considerably due to the Covid-19 spread.

Covid-19 originated in Wuhan, China during late December 2019, and was classified as a pandemic on March 10, 2020 (World Health Organization). Its diffusion has triggered a massive spike in uncertainty. Major uncertainties surround almost every aspect: the infectiousness, prevalence, and lethality of the virus; the capacity of healthcare systems to meet an extraordinary challenge; how long it will take to develop and deploy safe, effective vaccines; the duration and effectiveness of social distancing, market lockdowns, and other mitigation and containment strategies. Other socio-economic concerns refer to the near-term economic impact of the pandemic and policy responses, and specifically to the resilience of local systems as the pandemic recedes; whether "temporary" government interventions and policies will persist; the extent to which pandemic-induced shifts in consumer spending patterns will persist; and the impact on business survival, and all factors that affect productivity over the medium and long term (investments, R&D analysis [25]).

Italy had experienced the infection outbreak earlier and more consistently than other highly populated nations. The first case was announced on 21 February 2020 in Codogno, near Lodi and Milan, in the north of Italy.

In order to limit Covid-19 diffusion, the national and regional authorities have imposed social distancing measures and stay-at-home orders, including a ban on free movement except for the case of urgent needs. Due to these restrictions, people cannot move in other municipalities and they can go out of their home just for exceptional reasons, including food provision. Firms producing non-essential goods or services have been closed down [26]. These measures are having a large impact on employment, leading to a sharp rise in unemployment and other workers being given reduced hours or temporarily furloughed. Following the Prime Ministerial Decree of 22 March 2020 which forced all non-core or strategic production activities to close, it is estimated that 7.8 million workers are temporarily unemployed, for industry six out of 10 workers (59.6% of the workforce), while for services the block would affect just over a quarter of workers (26.7%).

This causes some uncomfortable situations. People living in municipalities where there are no supermarkets have difficulty in getting food. Many companies, especially in the tourism and hotel sector, are at risk of failure and this could lead to the loss of many jobs and the reduction of income for many families. Many workers are at risk of unemployment. In part of the population, the fear of contagion is spreading, and this leads to a drastic reduction in movements.

These three conditions (mobility restrictions, possible reductions in household income together with the fear of infection) make access to food more complicated.

Due to these difficulties in food accessing in this period, several self-organized food provision initiatives are emerging at municipal level to re-organize the distribution chain and connect people in need, volunteers, voluntary associations and small producers. These iniatives represent a favorable environment where social innovative experiences can be tested. They contribute to solve a current societal challenge, i.e., the recent problems in the food access (physical distance, restrictions of mobility and income barriers) through innovative organizational changes in service provision, which reconfigure existing social practices. Doing so, they attempt to enhance outcome and social well-being and improve the cooperation among population, local associations and service providers.

## 4. The Test Area

The article analyses self-organized food provision initiatives that emerged in Campania region and in particular the province of Naples, in the South of Italy, in March and April 2020 (Figure 1).

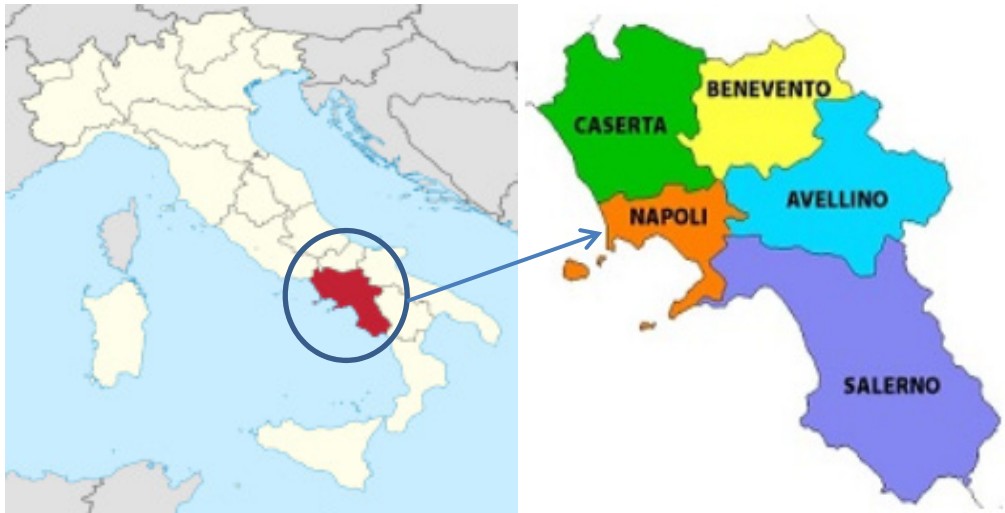

**Figure 1.** Location of Naples (Napoli) within Italy.

The South of Italy, a vast and densely populated region, has lagged behind the rest of the country, in spite of the big investments in development made over the last 60 years. Understanding the reasons for this failure is particularly important not only for the future of Italian citizens, but also for the development of regional policies in Europe [27]. There is of course a historic North–South dualism analysed in different studies both in socio-economic terms and in terms of perception [28–30].

The province of Naples has the highest number of infections in all of Southern Italy (2026 infections out 3 million of people at 16 April [31]) and it is implementing restricting measures to contain the virus diffusion (closure of all non-essential activities, restrictions on the movement of people). This province is located in a region (Campania) where 41.4% of the population is at risk of poverty (the highest level in the European Union [32]), i.e., 4 out of ten households have economic difficulties to access goods and services as their income is lower than the average Italian one. At regional level, in 2018, the employment fell, interrupting the expansive phase recorded from 2015 onwards. However, employment levels have maintained overall at 2008 pre-crisis levels. The unemployment rate remains high, despite the drop in 2018. The labour supply has contracted, reflecting the decline in both the number of job seekers and the number of people seeking employment and in that of the employed. The worsening of labour market conditions has affected household consumption, which has slowed down overall [33]. Campania has a higher-than-average spread of poverty than the Italian average and a wider inequality of income. The welfare of families in Campania suffers from high-income disparities, lower quality of public services and poorer health than in other areas of the country (ibid.).

Therefore, the restricting measures could have important effects on the already struggling local economic system, getting worse income disparities, economic downturn, and unemployment. This in turn could increase the number of people needing food assistance.

The province has recently experienced some social innovation initiatives (e.g., with San Gennaro Community Foundation, a non-profit organization founded by Don Antonio Loffredo that creates job and life opportunities for the young people of the Rione Sanità in Naples, a difficult reality but full of potential; or with Naples 2.0 social innovation incubator, etc.). Here these experiences have found fertile ground thanks to a strong vocation for volunteering, especially in the protection and promotion of the artistic heritage and the support of families in need.

The province has also accelerated the growth of citizen-led food system initiatives, above all for creating innovative and inclusive job opportunities at local level and redesigning urban-rural linkages among its territories. As such, it has demonstrated an increasing interest for alternative food networks and for practices and tools that foster multi-stakeholder approach.

## 5. Methods

The characterization of the self-organized initiatives for food provisioning in the urban-rural territories in the province of Naples takes place starting from the open-street map "Reti e pratiche solidali nell´emergenza" (Networks and solidarity practices in the emergency) [34]. In this map, all Neapolitan citizens post the self-organized initiatives emerging in the last two months (March and April 2020) to fight fragility, social exclusion and isolation, as well as to resolve daily problems becoming difficult to overcome due to the restrictions imposed to the Covid-19 containment measures. Among these initiatives, we consider specifically those related to the food provisioning and mapped at 10 April 2020. For each of them, we search for secondary data related to:

(1)    The innovative character, outlined by their capacity to reconfigure existing social practices
(2)    The existence of societal challenges to which the initiatives try to give a solution
(3)    The attempts to enhance outcome and social well-being underlined by their activities and aims
(4)    The engagement of society, i.e., the presence of society, which is not just the beneficiaries of the initiatives, but also the promoter and/or the protagonist of the initiatives.

These criteria are directly derived from the definition of social innovation developed within SIMRA project.

We have looked for this information firstly on the open-street map. At a later time, we have analysed documents related to the initiatives, such as statutory or project documents, newspapers, scientific papers, Facebook posts in fan pages or groups, and discussion forums. We collect these documents thanks to research on Google and Google Scholars for the period March–April 2020.

Subsequently, we describe the Masseria Ferraioli project. We have decided to analyse this project among those previously identified, mapped and validated according to the four criteria because it is a good example of social innovation as defined by SIMRA. In addition to the four criteria test, the case study has been further assessed by a semi-structured interview in order to achieve a (partial) validation of the secondary data analysis. This interview is composed of two parts. The first one is related to the history and general information about these initiatives (place, foundation year, activities, private and public actors involved, etc.). The second part aims at testing the correspondence of the initiatives to the characteristics of a social innovation experience, as delineated by the SIMRA project.

This project has drafted a list of check questions to test deeper the initiatives with strong social connotation. This list includes some questions, which help to define the characteristics of social innovation as process, as product and the relative output, along and in detail of the four above-mentioned pillars. For each of them, the project defines also the level of requirement. In other terms, if the level is recognized as necessary, the features that are mentioned in the question are an essential prerequisite of social innovation. An experience is configured as social innovative only in the case it answers positively to the majority of questions and satisfies the related level of requirement. Without these requirements, the initiative cannot be defined as social innovation experience. Table 1 reports the questions and the level of requirement. This table has been administered to the project manager and compiled together with him.

**Table 1.** The check questions list and the level of requirement (Source: SIMRA project, 2020).

| Check Question | Level of Requirement |
| --- | --- |
| Social innovation as process–Pillars 1-2-3 <br> Is there a process of reconfiguration of social practices (e.g., relationships/collaborations/networks/institutions/governance structures) in response to societal challenges | Necessary |
| Does the novelty/reconfiguration take place in new geographical settings or contexts, or in relation to previously disengaged social group(s)? | Necessary |
| Does the process of novel reconfiguration involve members of civil society as active participants? | Necessary |
| Does the process of reconfiguration result in new social practices that increase the engagement of civil society actors? | Possible but not necessary |
| Does the SI arise as a result of a crisis or apparently intractable problem? | Possible but not necessarily |
| Can a public agency be the initiator and/or driver of social innovation? | Possible but not necessarily |
| Can social innovation be initiated by a private sector agency? | Possible but not necessarily |
| Is the social innovation process driven by certain values and ethical positions? | Possible but not sufficient and context-dependent |
| *Social innovation as product–Pillar 4* <br> Do new social practices engage voluntarily civil society actors (in relationships/collaborations/networks/institutions/governance structures) as a result of the Social innovation? | Necessary |
| *Outcomes/Impacts arising from social innovation–Pillar 3* <br> Do these reconfigurations enhance outcomes on societal well-being, i.e., in relation to society, economy, environment or any combination thereof? | Desirable, but not necessarily happens |
| Are trade-offs between types of benefit or beneficiaries likely to arise as a result of social innovation? | Possible but not necessarily |

The Masseria Ferraioli project manager was also free to express the project idea, the mission and the vision of his initiative, etc. His statements are reported in this text.

According to Oxfam [35], the international confederation of non-profit organizations, the impact of the pandemic threatens to reset the progress made over the last 10 years in the fight against extreme poverty in some regions, the most deprived in the world. This is probably true also for the province of Naples that, as we mentioned before, is at risk of relative poverty. The reasons why we have decided to focus on Naples province are already listed in the introduction (low income, increasing interest for citizen-led food system and social innovation initiatives). In addition, we mention the high number of people who have already access to temporary assistance from Naples municipality in the last two months. 13,502 people have in fact benefited from the Municipal Solidarity Fund, a public–private fund that distributes vouchers for food purchasing to people in need. Finally, we are especially interested in local initiatives that reconfigure urban–rural linkages. Emergencies such as the spread of Covid-19 impose a territorial integrated approach to development which go beyond intra-city policy coordination and traditional rural issues, but promote the integration with surrounding areas, both urban and rural, reconfiguring their relations. This is crucial in terms of food provision. The benefits of stronger urban–rural cooperation include more efficient land use and planning, better provision of services (e.g., food provision, but also logistic and transport services) and better management of natural resources. This appears even more critical in the province of Naples, where the agriculture is the one of the most affected by the environmental crisis of the Campania Plain ("Land of fires"). This environmental crisis requires the need for specific support measures for farms affected by the crisis, in order to maintain the agricultural garrison, ensuring the highest level of safety for citizens (Law 6/2014 of the Campania Regional Plan.)

### 6. The Map of Food Provisioning Self-Organized Initiatives in the Urban-Rural Territories in Naples Provinces and Their Characterization according to the Four Pillars of Social Innovation

Food provisioning is a very important issue during the COVID 19 emergency. The province of Naples has activated several initiatives in an attempt to support the citizens in difficulty. These initiatives include those related to food and can be clustered into three macro-areas: food banks, canteens and other projects to support food provisioning (institutional projects) (Figure 2).

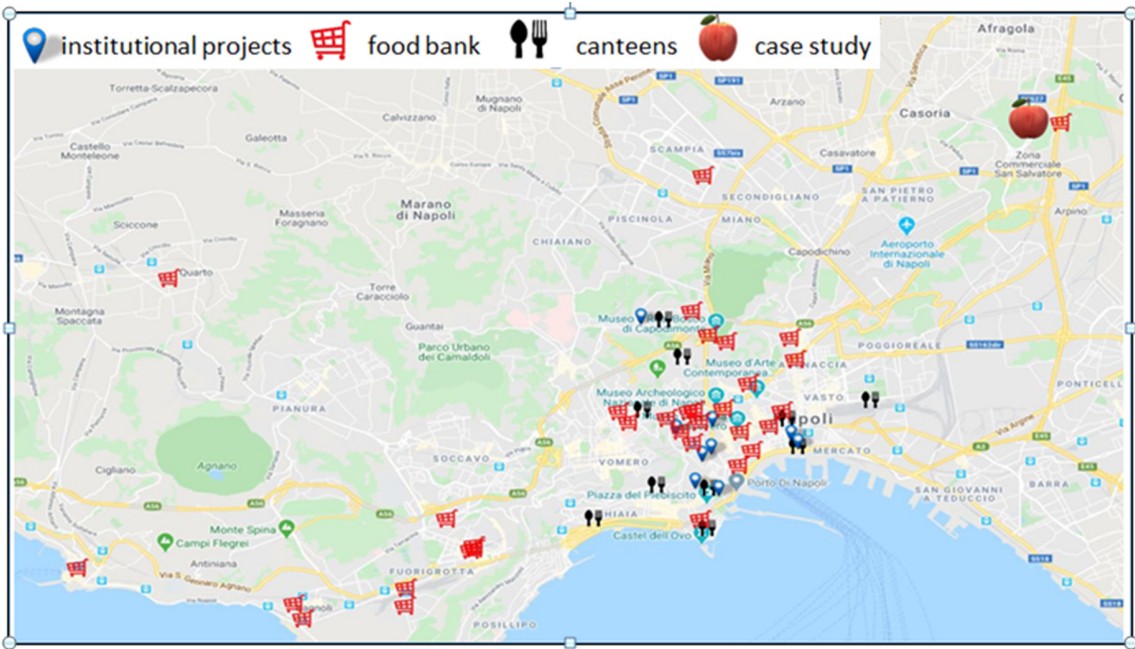

**Figure 2.** The map of food provisioning self-organized initiatives in the urban-rural territories in Naples provinces during the period of Covid 19. Source: Umap, 2020 (last access 10 April 2020).

The map takes into consideration 52 initiatives in support of food provisioning: 33 in reference to the food bank, 9 canteens and 10 strictly within the competence of the institutions.

We then analyse these initiatives in reference to the four pillars of the SIMRA social innovation framework. Table 2 lists all considered initiatives and includes the indication of the pillars possessed by each experience.

Among these initiatives, there are those related to the suspended spending (recovering the ancient Neapolitan custom of leaving an already-paid-for coffee at the bar for the people in need (Caffè sospeso—Suspended coffee)). Suspended spending enables to donate food, medicines and other basic necessities to those who are in a state of need leaving products at the checkout store. Donation is reported with a WhatsApp message or with a phone call at the same time as a fundraiser has been activated. The first institution to activate this form of solidarity spending in Naples is the II Municipality.

Among the promoters of these initiatives, there are also the Citizenship Agencies, which are a reference point for the elderly, disabled people or those ones at high risk of exclusion. These agencies are managed by networks formed by voluntary associations and third sector realities. During the Covid-19 emergency, they have delivered home shopping and medicines (in compliance with the rules of distancing), and offered telephone and remote support through Skype and YouTube.

The canteens continue working by distributing food bags to the destitute and homeless. The consumption of food no longer takes place in their spaces for the safety of operators and users.

**Table 2.** Initiatives in support of food provisioning and the coherence with the four pillars of social innovation definition by SIMRA.

| Initiatives in Support of Food Provisioning [1] | 4 Pillars of the SIMRA Social Innovation Framework [2] |
| --- | --- |
| Area Flegrea Solidale | 3,4 |
| Area Flegrea Solidale | 3,4 |
| Area Flegrea Solidale | 3,4 |
| BAM—Brigata di Appoggio Mutuo Napoli | 2,3,4 |
| Banco alimentare | 2,3 |
| Brigata di solidarietà Vincenzo Leone | 3,4 |
| Brigata di solidarietà Vincenzo Leone | 3,4 |
| Brigate di solidarietà Quarto (NA) | 3,4 |
| Durante l'emergenza attiviamo la solidarietà | 2,3,4 |
| Fuorigrotta solidale | 3,4 |
| IL MUTUO SOCCORSO AI TEMPI DEL COVID19 | 3,4 |
| IL MUTUO SOCCORSO AI TEMPI DEL COVID19 | 3,4 |
| IL MUTUO SOCCORSO AI TEMPI DEL COVID19 | 3,4 |
| IL MUTUO SOCCORSO AI TEMPI DEL COVID19 | 3,4 |
| IL MUTUO SOCCORSO AI TEMPI DEL COVID19 | 3,4 |
| JUST PANARO! TU RESTA A CASA, LA SPESA LA FACCIA | 3,4 |
| JUST PANARO! TU RESTA A CASA, LA SPESA LA FACCIA | 3,4 |
| JUST PANARO! TU RESTA A CASA, LA SPESA LA FACCIA | 3,4 |
| JUST PANARO! TU RESTA A CASA, LA SPESA LA FACCIA | 3,4 |
| JUST PANARO! TU RESTA A CASA, LA SPESA LA FACCIA | 3,4 |
| JUST PANARO! TU RESTA A CASA, LA SPESA LA FACCIA | 3,4 |
| JUST PANARO! TU RESTA A CASA, LA SPESA LA FACCIA | 3,4 |
| JUST PANARO! TU RESTA A CASA, LA SPESA LA FACCIA | 3,4 |
| MASSERIA ANTONIO ESPOSITO FERRAIOLI | 1,2,3,4 |
| NAPOLI SOLIDALE | 3,4 |
| NAPOLI SOLIDALE | 3,4 |
| NAPOLI SOLIDALE | 3,4 |
| NAPOLI SOLIDALE | 3,4 |
| NAPOLI SOLIDALE | 3,4 |
| NAPOLI SOLIDALE | 3,4 |
| Porta-spesa solidale a Montesanto/Quartieri Spagnoli | 2,3,4 |
| Responsabilità Popolare - Emergenza Senza Dimora Nap | 3,4 |
| Spesa solidale zona via Salvator Rosa | 3,4 |
| Binario della solidarietà | 1,3,4 |
| Chiesa S. M. Apparente | 1,3,4 |
| Drop-in Dedalus | 1,3,4 |
| Gruppo mense CENTRO LA PALMA | 1,3,4 |
| Mensa del Carmine | 1,3,4 |
| Mensa dell'Arco Mirelli | 1,3,4 |
| Mensa S. Brigida | 1,3,4 |
| Mensa S.Antonio di Padova ( via dei Pini) | 1,3,4 |
| Mensa S.Lucia | 1,3,4 |
| Parrocchia S. Maria del Buonconsiglio a Confalone | 1,3,4 |
| CARRELLO SOLIDALE | 3,4 |
| InsiemeMaiSoli-Comune di Napoli | 1,3,4 |
| SPESA SOSPESA-Municipalità 2 del Comune di Napoli | 2,3,4 |
| SPESA SOSPESA-Municipalità 2 del Comune di Napoli | 2,3,4 |
| SPESA SOSPESA-Municipalità 2 del Comune di Napoli | 2,3,4 |
| SPESA SOSPESA-Municipalità 2 del Comune di Napoli | 2,3,4 |
| SPESA SOSPESA-Municipalità 2 del Comune di Napoli | 2,3,4 |
| SPESA SOSPESA-Municipalità 2 del Comune di Napoli | 2,3,4 |
| SPESA SOSPESA-Municipalità 2 del Comune di Napoli | 2,3,4 |

[1] In some cases, the same experience is repeated in different areas. [2] The four pillars are: (1) the reconfiguration of social practices, (2) the existence of societal challenges to whom social innovation initiatives try to give a solution, (3) the attempts to enhance outcome and social well-being, (4) the engagement of society.

### 7. The Case: The Masseria Ferraioli

The Masseria Ferraioli is located in a property confiscated from the Camorra about 20 years ago. After many years of neglect, the municipality of Afragola has adopted a regulation for the allocation of the confiscated estate and started assignment to volunteer associations. The farm is dedicated to the memory of Antonio Esposito Ferraioli, innocent victim of the Camorra killed in 1978, and has been assigned to a network of five partners with different roles and skills:

- Consortium of Social Cooperatives "Third Sector"
- The volunteer association "Sott'e'ncoppa"
- Naples Metropolitan Chamber of Labour
- The social cooperative "L'uomo e il legno"
- The Cooperative of Giancarlo Siani

The Consortium of Social Cooperatives "Terzo settore" is the leader of the project, and since its foundation under Law 381/91 in 2002, has operated to promote social innovation initiatives through new methodologies and technologies. The voluntary association Sott'e'ncoppa was born in 1999 from the need to have an innovative social gathering space using the Fair Trade channel promoting Critical Consumption to raise awareness of social and environmental issues. Naples Metropolitan Chamber of Labour is the CGIL's metropolitan territory articulation. Founded in 1894, the CGIL of Naples has always fought for the rights of workers and against the Camorra. Over sixty trade unionists have lost their lives at the hands of the Mafia, and their names are read every year on 21 March, on the Day of Remembrance of the victims of the Camorra. The social cooperative "L'uomo e il legno" was founded about 20 years ago and deals with various and different activities including training, social farming and job placement. For the Covid-19 emergency, it has created a canteen for the distribution of meals. The Cooperative of Giancarlo Siani has created a Radio to support social redemption, and commitment in favor of the weaker groups, minors. In addition, it has an agricultural and beekeeping project, curated by professionals from the Faculty of Agriculture of the University Federico II of Naples.

The project of Masseria consists in the recovery of the old farm in order to create a space for educational and practical workshops on food issues, a shop for the sale of agricultural products from the Masseria and other farms, and a refreshment point, with the specific aim to give job opportunities to the poor people. In the next months, a part of the farm will be used as a home for women victims of abuse, which will work in agro food activities. The land near the farm is 12 hectares, one of which is intended for community gardens and therefore divided into more than 128 plots. Among the numerous cultivations, the most important are the orchard of about 1200 peach trees, ancient Sicilian wheat, potatoes to create a bread with potatoes typical of the area and "friarielli" (turnip tops) to prepare a pesto sauce. The sale of these products takes place through joint purchasing groups and in the future with online sales. The orchard has an extension of about six hectares. In addition to a productive role for the Solidarity Purchasing Group and the processing laboratory, it is designed to recover native but now abandoned varieties and create a museum of biodiversity of agricultural species in the area. With the aim of recreating a piece of the typical landscape of the Campania Plain, the orchard is also structured as an innovative museum through a "path of memory". The women of the social enterprise will organize and manage the visits to the orchard.

Firms wishing to use the farm will have to sign a policy document with precise constraints linked to working conditions and product quality. The production plant will have a processing capacity of 1000 liters per hour. In addition to the usual products such as tomato puree, ready-made sauces and in oil, it will be used to produce the "boccaccielli" through the technique of the cooking pot (boccaccielli are small jars that contain tomato puree or other creamy delicacy to be enjoyed). The idea is to bring the dishes of the Neapolitan culinary tradition to the customers with an optimal conservation. The regulation of the community gardens has been formulated at the end of a participatory process open to all interested public and private actors. It does not provide for any particular discriminating features with regard to the target audience or territorial origin. On the contrary, it has stringent

prohibitions on the consumption of plastic. These plots are rented through an annual concession, against the payment of a reduced fee of five euros per month. The land and water have been previously analysed to verify their quality and their possible use for production purposes. Relative tests have had positive responses. The gardens management promotes the possibility to spend own free time in a recreational activity and it contributes to social inclusion and educational growth of the local community through inter-generational exchanges aimed at promoting the excellence of the territory and local biodiversity. During the Covid-19 emergency period, urban gardens were not accessible and could not be grown according to national decree. Many petitions of environmental associations have been submitted for a derogation from this prohibition since the risk of compromising the summer harvest was high. Some Italian regions such as Lazio and Sardinia have made special laws to ensure the possibility of continuing such activities. Masseria Ferraioli has not stopped its activities and has joined the various initiatives to support the food provisioning of the provincial area of Naples.

Together with the civil protection and the municipality of Afragola, the Masseria has donated 36 boxes of food to the parishes of St. Anthony and St. Michael, "so that they could be distributed to the neediest". According to the farm manager, "thanks to our civic sense, we continue to respect the measures taken by the competent authorities and indicated by experts to limit the risk of infection, with enormous sacrifice." The coronavirus epidemic will increase the difficulty of access to food, which already represent a real emergency", explains the manager. An important role will have to be played by the social economy, which finds its natural place in the goods confiscated from crime in keeping urban gardens closed. By 2021, the manager of the Masseria will consider completed the renovation work on the property (Figures 3–5).

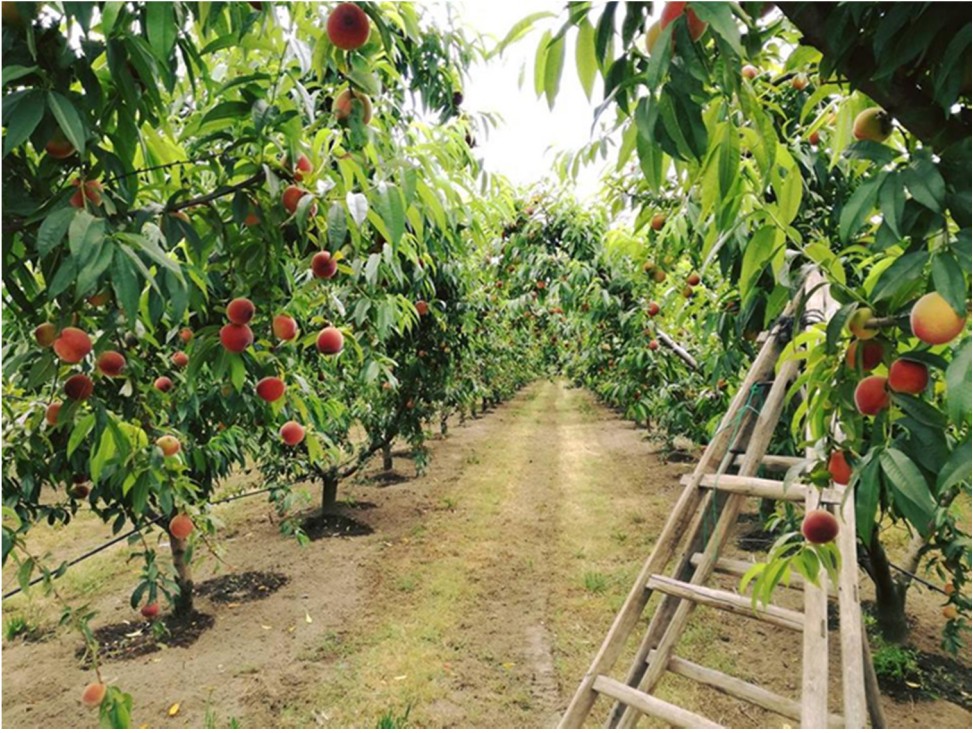

**Figure 3.** Peaches at Masseria Ferraioli, 2020.

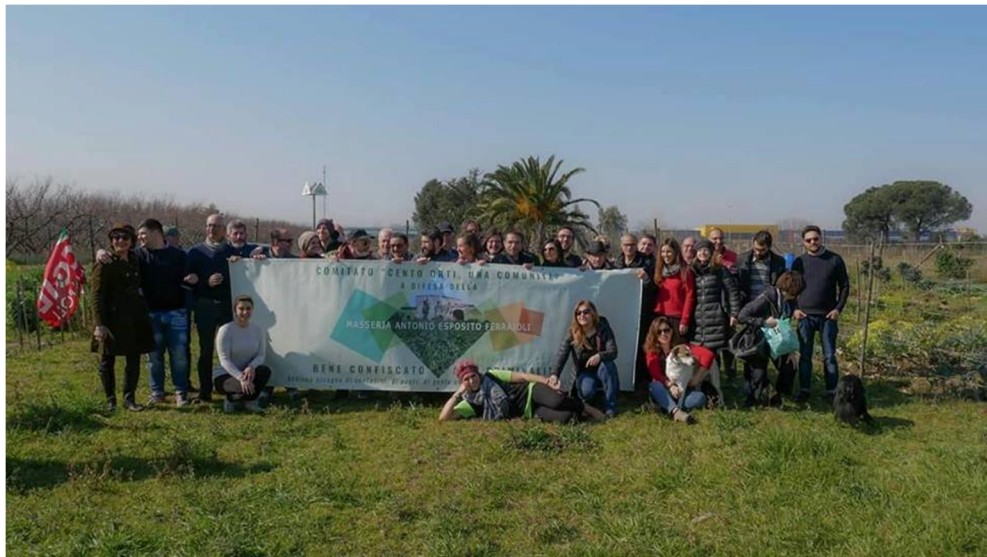

**Figure 4.** The volunteers of Masseria Ferraioli, 2020.

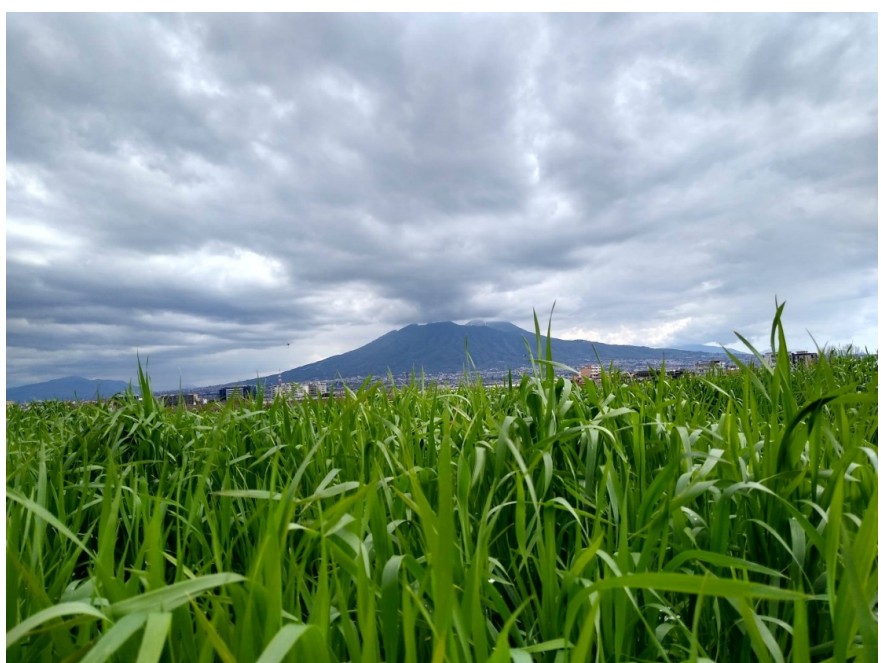

**Figure 5.** The view of Vesuvius from one of the fields of the farm.

## 8. Discussion

As we can see from Table 2, the 52 initiatives in support of food provisioning analysed here respond simultaneously to several pillars of social innovation. The largest group (29 initiatives) consists of the initiatives that enhance outcome, social well-being and the engagement of society (pillars 3,4) and results decisive in designing and fostering long-term thinking and sustainable production and consumption styles, as well as corporate and community well-being [36]. Eleven initiatives are in support of the reconfiguration of social practices as well as enhancing outcome, social well-being and the engagement of society. Ten initiatives are in support of Pillars 2, 3 and 4 of social innovation. In those contexts, we detect that new research flows concerning sustainable development no longer refer exclusively to the environmental sphere, but they also reflect multidimensionality of the international development agenda, in line with human and local priorities [37]. Only one initiative (Banco Alimentare) is characterized by a combination of the existence of societal challenges—to which social innovation

initiatives try to give a solution—and the attempts to enhance outcome and social well-being. Finally, only Masseria Ferraioli responds to all the pillars of social innovation. With reference to this initiative, the results emerging from the interview with the farm manager Giovanni Russo give very positive results. The interview is divided into three phases. In a first phase, we focus on the birth of the project and the initiatives that over the years have been carried out. The second part of the interview analyses the relationship with social innovation by focusing on the questionnaire relating to the question list and the level of requirements (Table 1) In this context, the manager states that the project responds to social innovation through two cornerstones: training and training laboratory, and recovering the ancient traditions and crops. The manager says: "It is happening, not without difficulty, a reconfiguration of the territory thanks to very uneven groups, who collaborate and confront each other from the old farmer who rediscovers the passion for the land, to young students now representatives of the law enforcement. Finally, during the third phase the Covid problem is tackled in relation to the activities, which highlight the need to restart in order not to lose the spring crops and to give gardeners the possibility to access their own spaces and socialize." The continuous references to the valorization of the territory highlight how the project wants to contribute to its requalification and to be part of the reconfiguration of the relationship between town and country.

Analysing Table 1, we notice that all the necessary requirements are met and most of those possible. There is a reconfiguration process of well-articulated social practices in response to the challenges of society. This is highlighted by the various projects within the farm and the change in the distribution food chain to stem the new agri-food problems related to the Covid-19 emergency.

The territorial reconfiguration takes place in a geographical context where disengaged social groups live. This is even clearer in relation to community gardens that are managed by people with different socioeconomic backgrounds and ages. The reconfiguration process involves the stakeholders in an active way. Once a month there are meetings for participatory management of the farm; social networks are used by users to submit proposals or critical issues. Effective territorial regeneration is only possible through the satisfaction of the different stakeholders´ needs: the municipal administration, the gardeners and the partners that act as an intermediary [38]. The new social practices activated in this project confirm its social innovative characterization: these include the numerous cultivation techniques used, the different distribution channels of their products up to the desire to achieve online sales. Additionally, this is also underlined by the activities carried out within the community gardens, which contribute to the creation of a model of agriculture for the home markets that arise from the idea of "gift economy" [39].

## 9. Conclusions

The paper succeeds in combining both social innovation policies and the changes needed to support the Covid-19.

Specifically, it focuses on the self-organized initiatives carried out in support of food provisioning in urban–rural territories in the province of Naples. By order, it offers a first map of these initiatives and their clusterization into three macro-areas: food banks, canteens and other projects to support food provisioning. This exercise attests the initiatives' unbalanced spatial distribution across Neapolitans urban–rural territories, as they are located above all in the central districts. Secondly, the paper includes a brief description of these initiatives according to the social innovation pillars. This activity enables to validate their characteristics, and specifically their propensity to improve the outcome and social welfare as well as to involve local society. Results suggest that all initiatives have all the typical requirements of social innovation theorized by SIMRA; however, none of these can be defined exactly as a social innovation experience according to the definition of this project. Masseria is an exception: it emerges as a social innovative best practice as it presents some features that satisfy all four pillars.

The current study highlights also the importance of the combined commitment of local communities and volunteering association, as well as an increasing interest in reconnecting people with food practices. This appears as a priority above all in this period when food access has become more difficult.

In the future, more attention should be paid on the capacity of all mapped initiatives to satisfy local food needs, as well as the ability to transform local social practices. Quantitative and sociological studies should be implemented accordingly. Similarly, future investigation should analyze each project in detail and test their consistency with the four pillars of social innovation through detailed interviews. The overall validation of all considered self-organized initiatives like that realized for Masseria through semi-structured interview was impossible at this time due to the current critical situation. Finally, as this paper refers just to the initiatives developed during the first two months of the Covid-19 emergency, it would be interesting verify the development of new initiatives or the permanence of those already mapped in the next months.

**Author Contributions:** Conceptualization: V.C., V.R.; Data curation: V.R.; Formal analysis: V.C., V.R.; Investigation: V.C., V.R.; Methodology: V.C., V.R., Validation: V.C., V.R.; Writing—original draft: V.C., V.R.; Writing: review & editing: V.C., V.R. All authors have read and agreed to the published version of the manuscript.

**Funding:** No external funds.

**Acknowledgments:** The authors thank Giovanni Russo, Project Manager of Masseria Ferraioli, for his availability. The authors thank the Department of Innovation, Research and University of the Autonomous Province of Bozen/Bolzano for covering the Open Access publication costs.

**Conflicts of Interest:** No conflicts of interest.

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
