# Peer review of "Social Innovation and Food Provisioning during Covid-19: The Case of Urban–Rural Initiatives in the Province of Naples"

_sustainability, doi:10.3390/su12114444_

Round 1

Reviewer 1 Report

The proposed work is a relevant exercise in the field of food sustainability, focusing on food provisioning and social innovation in the Global North. It presents interesting elements that must be taken into account amidst the Covid-19 emergency in relation to the issue of food provisioning and the opportunities to create novel social innovation proposals – the latter also representing a strategy to tackle criminal activities and the role that can be played by the civil society. The manuscript is of fit for the journal’s scope and its research question is enthralling and up-to-date. The paper structure and the organisation of its contents are convincing.

The case study results appropriate to the scope of the present paper. The manuscript demonstrates a sufficient understanding of the relevant literature in the theme of SI, although more information about Covid-19 is needed in order to understand the evolution and the connection between food provisioning and SI. The literature explored is recent, the references used are of interest and meet the journal’s requirements. More specifically, the social innovation side seems to be covered. The authors might want to enhance their explorations on the food-criminality nexus (in Campania), as well as rural/urban development, food/resource sustainability, wellbeing and related topics. Perhaps the following publications might be of help?

  • Agovino, M., Cerciello, M., & Gatto, A. (2018). Policy efficiency in the field of food sustainability. The adjusted food agriculture and nutrition index. Journal of environmental management, 218, 220-233.
  • Gatto, A. (2020). A pluralistic approach to economic and business sustainability: A critical meta-synthesis of foundations, metrics, and evidence of human and local development. Corporate Social Responsibility and Environmental Management, 2020. https://doi.org/10.1002/csr.1912.
  • Gatto, A., Polselli, N., & Bloom, G. (2016). Empowering gender equality through rural development: Rural markets and micro-finance in Kyrgyzstan. L’Europa e la Comunità Internazionale Difronte alle Sfide dello Sviluppo, 65-89.
  • Giordano, A., & Tarro, G. (2012). Campania, terra di veleni.
  • Morrow, N., Salvati, L., Colantoni, A., & Mock, N. (2018). Rooting the Future; On-Farm Trees’ Contribution to Household Energy Security and Asset Creation as a Resilient Development Pathway—Evidence from a 20-Year Panel in Rural Ethiopia. Sustainability, 10(12), 4716.

The results clearly show the consequences of the issue. The decisions taken by the support system about the compliance of food provisioning requirements are foreseen within different scenarios from a long-term point of view. The methodology seems to be appropriate, but it could be better clarified. Although the interviews’ questions are clearly sketched, the interviewing process shall be exposed more detailly. In the conclusion section, it would be good to better frame this study’s limitations and future research – that was, indeed, outlined. I also suggest a linguistic revision – some typos and syntax errors appear here and there. Some acronyms should be spelt at their first use.

Provided that the corrections for these minor issues will be addressed, I advise the publication of the manuscript on Susy.

Reviewer 2 Report

This paper is extremely interesting, as it casts light on social innovation in Southern Italy, in particular in Naples, one of the most populated Southern metropolitan areas, characterised by several social, institutional and economic problems. The structure of the paper is fine, objectives are clear, analysis is well conducted and presented, and at the end, authors are able to response to the research questions. However, I suggest some revisions:

- First, I have to point out that English language is not perfect. The paper would require an English language revision.

- I think that the case study area should be described and framed, I mean contextualised better. In the introduction, when authors present the case of Naples, they should talk about the macro-regional context, I mean the question of the Mezzogiorno. So, the reader can understand better the historical, geographical, institutional and socio-economic macro-context that explains better why Naples is in this socio-economic condition. Authors could add a sub-paragraph in the introduction, a bit of lines, where they explain precisely that the socio-economic development of Naples can be better understood in the context of the “Questione Meridionale”, and the North-South divide. In this respect, you can refer to the huge relevant scientific literature (for example: Cannari and Franco, Il Mezzogiorno: ritardi, qualità dei servizi pubblici, politiche; Daniele and Malanima, Il prodotto delle regioni e il divario Nord-Sud in Italia (1861–2004); Felice, Perché il Sud è rimasto indietro; Vecchi, In ricchezza e povertà. Il benessere degli Italiani, dall’Unità ad oggi; Musolino, North-South divide in Italy: reality or perception?; Svimez, 150 anni di statistiche italiane Nord e Sud 1961–2011; Wolleb e Wolleb, Divari regionali e dualismo economico)

- As far as the better understanding of the case study area is concerned, I would even suggest, if possible, to insert a map of Italy where you highlight the area of Naples.

- If possible, authors should give a little more value to the interview, for example using a little bit more the same words – quotations - of the interviewee (manager of Masseria), in order to provide clear evidence about his ideas and his point of view. Anyway, at least they should explain better how they used the interview, as it is not clear enough (I suppose that in the description of what Masseria Ferraioli does authors used the content of the interview as well). 

- Moreover, I would suggest to make some references to the literature concerning urban-rural linkages. Authors mention frequently this issue, but without making any reference to the literature. See for example the several works by Eveline van Leeuwen. This is a topic rather important in the literature about regional, and rural, development.
